# Triplexes Color the Chromaverse by Modulating Nucleosome Phasing and Anchoring Chromatin Condensates

**DOI:** 10.3390/ijms26094032

**Published:** 2025-04-24

**Authors:** Alan Herbert

**Affiliations:** Discovery, InsideOutBio, Charlestown, MA 02129, USA; alan.herbert@insideoutbio.com

**Keywords:** Triplex, G-quadruplex, Z-DNA, flipons, condensates, gene expression, nucleosome phasing, endogenous retroelements, embryonic development, helicase, microRNA, lncRNA, anti-sense RNA, argonaute, enhancer, promoter

## Abstract

Genomic sequences that form three-stranded triplexes (TPXs) under physiological conditions (called T-flipons) play an important role in defining DNA nucleosome-free regions (NFRs). Within these NFRs, other flipon types can cycle conformations to actuate gene expression. The transcripts read from the NFR form condensates that engage proteins and small RNAs. The helicases bound then trigger RNA polymerase release by dissociating the 7SK ribonucleoprotein. The TPXs formed usually incorporate RNA as the third strand. TPXs made only from DNA arise mostly during DNA replication. Many small RNA types (sRNAs) and long noncoding (lncRNA) can direct TPX formation. TPXs made with circular RNAs have greater stability and specificity than those formed with linear RNAs. LncRNAs can affect local gene expression through TPX formation and transcriptional interference. The condensates seeded by lncRNAs are updated by feedback loops involving proteins and noncoding RNAs from the genes they regulate. Some lncRNAs also target distant loci in a sequence-specific manner. Overall, lncRNAs can rapidly evolve by adding or subtracting sequence motifs that modify the condensates they nucleate. LncRNAs show less sequence conservation than protein-coding sequences. TPXs formed by lncRNAs and sRNAs help place nucleosomes to restrict endogenous retroelement (ERE) expression. The silencing of EREs starts early in embryogenesis and is essential for bootstrapping development. Once the system is set, EREs play a different role, with a notable enrichment of Short Interspersed Nuclear Repeats (SINEs) in Enhancer–Promoter condensates. The highly programmable TPX-dependent processes create a chromaverse capable of many complexities.

## 1. Introduction

Genetic elements, called flipons, adopt alternative structures to produce different biological outcomes. They encode information by their shape rather than by their sequence. By switching states from one conformation to another, flipons change the readout of RNA from the genome. The transition from one flipon structure to another occurs without strand breakage or any alteration to the nucleic acid sequence.

Flipons are named for the alternative conformation they encode, e.g., Z-flipons form Z-DNA and Z-RNA, G-flipons form DNA and RNA G-quadruplexes (GQs), and T-flipons encode triplexes (TPXs). The formation of these alternative conformations inside cells is now well supported for both Z- and G-flipons. Here, the focus is on three-stranded TPX (Figure 1). In nondividing cells, the extra strand is usually composed of RNA, rather than DNA. The third strand can be encoded in regions quite distant from where the TPX forms. This design is quite variable and helps integrate the readout of genetic information in cells by establishing open regions of chromatin that are depleted of nucleosomes (historically called nucleosome-free regions (NFRs)). The flipons and transcription binding sites within NFRs enable the promoter–enhancer contacts that modulate gene expression. At these locations, an RNA cloud composed of promoter and enhancer transcripts engages RNA and protein effectors that set the chromatin state. The process is dynamic with both DNA and RNA conformations reset as an RNA polymerase is released from the promoter. This cycle allows an update of the promoter based on the current cellular condition.

## 2. Genomes Encode Information by Both Shape and Sequence

The human genome encodes many possible outcomes. Initially, chromosomes were described as composed of centromeres, telomeres, and chromomeres [1]. The work on telomeres and centromeres revealed that they were composed of repetitive sequences, which play a role in chromosomal maintenance during cell division and differentiation. As technology advanced, each chromomere region was assigned a color as its functionality was defined. The labeling was likely a play on the name chromosome that was coined in 1888 by Wilhelm von Waldeyer-Hartzchroma. He used the Greek word “χρῶμα” (khrōma), meaning “color”, as the root for the name. In these early maps of the chromaverse, heterochromatin was labeled black, and regions with actively transcribed genes were called red. Each domain within the nucleus was further shown to have its own location and unique sets of epigenetic marks [2]. High-resolution techniques, such as DNase I footprinting and ATAC-seq, then enabled the mapping of NFRs in each domain. These and other approaches identified the proteins bound, the sequence motifs recognized, and the functions performed. The work revealed the complexity of the chromaverse, demonstrating that the collage of colors within the composite maps was not fully defined by the linear flow of information from DNA to RNA to protein [3].

The ENCODE (Encyclopedia of DNA Elements of Human and Mouse Genomes) surprisingly revealed that many genomic regions within chromomeres were transcribed to produce non-protein-coding RNAs (historically referred to as ncRNAs). However, agreeing on the extent of this promiscuous expression took some time [4,5,6]. The current consensus is that over 75% of the human genome is read out as RNA, with 37,595 non-protein coding genes (ncRNAs) annotated compared to 20,225 transcripts that incorporate codons [7,8,9].

Recent high-throughput approaches have focused on ncRNAs and generated a wealth of new information. The ncRNAs vary in size, origin, copy numbers, protein binding partners, access to the nucleus, chromatin interactions, and even in definition [10,11,12]. At one end of this scale lie long ncRNA (lncRNAs) with low expression. Initially, lncRNAs were defined as longer than 200 nucleotides (nts) in length. Now it is suggested that 500 nts is a more appropriate size. The newer definition of small RNAs (sRNAs) includes a repetitive sequence like the Alu element (AE) repeat family of Small Interspersed Nuclear Elements (SINEs) and other transcripts such as microRNAs (miRNAs), piwi RNAs (piRNAs), transfer RNAs (tRNAs), ribosomal 5S rRNA, *7SK*, *7SL*, and vault and Y RNAs that with sizes in the 50–500 nt range [11].

Both tRNAs and miRNAs have been well characterized. Both are noncoding RNAs that impact mRNA translation into protein. The interactions of miRNA with transcripts are mediated by the argonaute (AGO) family of proteins, which also includes the PIWI clade that binds piRNAs [13,14]. In humans, there are four AGO proteins. AGO2 has nuclease activity and slices mRNA that are fully matched to the miRNA. AGO3 has slicer activity that is most evident with miRNAs of ~14 nts [15]. AGO1 and AGO4 are not known to have slicer activity. Interestingly, AGO proteins can engage tRNA fragments and other small RNAs to perform similar functions to those involving miRNAs [12,16,17]. Understanding of these functions has grown as well. Initially, the pathways were thought of as only cytoplasmic. They targeted RNA for elimination or inhibited its translation. Then, the involvement of miRNAs and piRNAs in gene suppression within the nucleus was revealed in a number of model organisms [18]. More recently, other sRNA pathways have been identified that lead to gene activation [19,20].

With the exception of miRNAs and tRNAs, other roles for the noncoding part of the genome were not explored or were considered unlikely, as argued by Brenner [21]. Most notably, the low complexity repeat sequences that were not within codons were considered uninformative due to their high frequency in the genome. Furthermore, early models proposing a role for repeats in tissue specification were not experimentally verified (Britten, 1969 #303 [22]). Additionally, the repetitive sequences were considered detrimental because of the enhanced error rate associated with their replication and their highly recombinogenic nature [23]. While these events increased genetic diversity and sometimes produced advantageous outcomes retained by genomes through natural selection, they also increased disease risk. At best, the persistence of these repetitive sequences in the genome was considered as one of nature’s quirky imperfections [21].

Quite separately, the ability of certain DNA sequences to form conformations other than the right-handed Watson–Crick B-DNA helix became apparent. Each structure was favored by a particular nucleotide motif, with polypurines favoring a three-stranded right-handed TPX [24,25,26] and guanine repeats favoring a four-stranded quadruplex composed of G-tetrads (GQs) held together by a combination of Watson–Crick and Hoogsteen hydrogen bonding schemes [27,28]. An alternating purine/pyrimidine motif favored left-handed Z-DNA formation [29,30,31], while a pair of cytosines could intercalate with another pair to form a four-stranded i-motif structure [32,33]. The sequences that form both B-DNA and these alternative conformations in vivo are called flipons [34]. They change biological outcomes by binding chromatin complexes that are specific for one conformation or the other [35]. Flipons are distributed nonrandomly throughout the genome. Recent evidence supports their important informational role in programming cellular responses.

G-flipons and Z-flipons have key roles in transcription and immunity. Each flipon type affects different outcomes. G-flipons regulate the onset of transcription by cycling between states. The complexes that they form remain locked and loaded until a particular event triggers the release of RNA polymerases from the pause position just downstream of the transcription start site. G-flipons also modulate many other processes, including replication and DNA repair [36,37,38]. Z-flipons play an important role in controlling innate immune responses to viruses and endogenous retroelements (EREs) that comprise autonomous Long Interspersed Nuclear Elements (LINEs), Long Terminal Repeats (LTRs), and non-autonomous SINE retrotransposons [39,40,41,42]. They also play a pivotal role in resetting promoters so that they are ready for the next round of transcription [43,44,45,46,47,48,49,50]. In addition to encoding Z- and G-flipons, EREs also form TPXs [51,52]. Most notably, SINEs represent the ultimate abstraction of genetic encoding by shape rather than sequence, packing many different flipon combinations into their 280 nt span, while not coding for any protein at all. Despite their small size, the genetic regulatory elements that they encode greatly impact a cell’s molecular biology.

## 3. T-Flipons and TPX Forming Sequences

The biology of T-flipons is quite different from that of G- and Z-flipons. Only in the case of mirror repeats, in which a sequence is repeated with its order reversed (e.g., “ACG” becomes “GCA”), can TPXs form locally. Otherwise, the third strand is sourced from a distant site, with both sRNAs and lncRNAs involved in the fold. These interactions can approximate DNA segments that are separated by long distances or transcribed from different chromosomes. The bridges between these sites are strengthened by the many RNA copies transcribed from each locus, creating an RNA that concentrates protein into condensates, some of which undergo phase transitions to fulfill and facilitate particular functions, from modulating gene expression to coordinating cellular responses during differentiation. To understand these processes, we will first discuss the sequences prone to form TPXs.

## 4. DNA and Hybrid TPX

Right-handed DNA duplexes can form TPXs with either a DNA or RNA third strand (Figure 1A). The TPX stability arises from the additional stacking interactions and the increased number of hydrogen bonds formed between bases compared to B-DNA. TPXs can also form with RNA inserted into either the major or minor groove. Major groove TPXs are most common (Figure 1A). Experimental studies have revealed several factors that influence the stability of major groove TPXs. They exist in equilibrium with other folds, including DNA:DNA (D:D) and RNA:RNA (R:R) duplexes and RNA:DNA hybrids (D:R). They compete with R-loops, which form when a DNA strand is displaced from the double helix by the binding of an RNA complementary to the other strand [53,54,55]. The different outcomes are summarized in Figure 1B. Double helices (either R:D or R:R) with an RNA purine strand are more stable than helices with a DNA purine strand and will preferentially remain as dsRNA or form an R-loop instead of a TPX. In TPXs, RNA pyrimidine strands bind to D:D, R:R, and D:R duplexes, whereas a DNA pyrimidine strand will bind only to duplexes that have a DNA, but not an RNA, purine strand. Hence, TPX formed with RNA and a D:D are the most stable of all the possible combinations.

The formation of triplets in vitro requires high levels of divalent cations to shield the charge repulsion arising from the approximation of the phosphate backbones forming each strand. The repulsion can also be countered by lowering the pH to protonate the cytosines in a polypyrimidine third strand, promoting the formation of an additional hydrogen bond in each triplet. In contrast, the hydrogen bonding scheme in the major groove of long d(A)_n_ repeat disfavors TPX formation. The bifurcated hydrogen bonds produce a high propeller twist of the bases, stabilizing a very rigid duplex structure. The zipper that forms can be disrupted by the insertion of other bases, enabling formation of a mixed sequence TPX with high thermal stability [56].

DNA-only TPXs arise from sequences with a mirror repeat (MR), in which the nucleotide order is reversed relative to the preceding repeat. The arrangement allows one of the repeats to fold back and hydrogen bond with a neighboring duplex region when the MR is 12 or more nts in length (Figure 1C). The formation of a TPX yields either an unpaired purine or pyrimidine DNA strand. A TPX incorporating the pyrimidine strand is favored under acidic conditions when the cytosine residues are protonated. TPXs formed in this way are referred to as H-DNA. They predispose to DNA damage and disease by stalling polymerases during cell division as the DNA strands are separated from each to undergo replication [26]. Over 85% of MRs in the genome are also perfect direct tandem homopurine repeats [57].

The “GT” TPX is another TPX variation with only G, T, or U in the third strand (Figure 1D–F). This motif is called “mixed” as the third strand can bind in either orientation to the duplex (Figure 1D). This dual outcome is unique to this TPX type. The orientation of the third strand depends on the duplex sequence. As with a standard TPX, the polypyrimidine third strand can have the same 5′ to 3′ polarity as the purine strand of the duplex. This “GT” parallel third strand forms with Hoogsteen bonds (Figure 1E). In contrast, a polypyrimidine third strand may form a noncanonical TPX by binding in an anti-parallel orientation relative to the duplex purine strand. Like a standard anti-parallel purine helix, the third strand is stabilized by reverse Hoogsteen bonds (Figure 1G). Regardless of the orientation, the third strand can contain a mix of pyrimidine and purine residues that form G:C*T and A:T*T but not A:T*A triads (where “*” indicates Hoogsteen base pairing of the third strand and “:“ a Watson-Crick hydrogen bond) [58]. By replacing “C” with “T”, the design overcomes the pH-dependence of “C” protonation necessary to stabilize the TPX. Substitution 2—aminopyridine or 5-methylcytosine—also increases TPX stability [59].

Third strands composed only of ”TCG” bases have also been explored as a way to target duplexes with a G repeat. In these TPXs, the strand orientation is parallel, with C:G*C^+^ triplets replaced by G:C*G. In contrast, a third strand composed of only ”GAU” bases docks in the anti-parallel direction with A:T*U, A:T*A, and G:C*G triplets [60]. Other studies examined third strands that pair through noncanonical triplets. Examples of these triplets are shown in Figure 2A. Up to two noncanonical triplet substitutions, such as G:C*A, G:C*U, and A:U*C, are tolerated in d(T:A)*rU TPXs longer than 19 nucleotides. The results reveal that some sequence variations in the RNA third strand are tolerated and confirm results from earlier reports [61,62]. Examples of such noncanonical base pairing in TPXs formed by ribosomal DNAs with their functional outcome are shown in Figure 2B and 2C and are discussed below.

There are many other base-pairing geometries compatible with triplet formation. A recent survey identified 108 predicted base triple families that belonged to 18 superfamilies. Of these, evidence for 68 was found in RNA structural databases [63]. The results suggest greater flexibility in triplet motifs than previously recognized. However, the number of these triples that can also assemble into stable triple-stranded helices is currently unknown.

## 5. Flipons That Fold in Many Different Ways

Like other flipons, TPX-forming sequences can fold into other alternative conformations. Some MR-like d(GCG GCG) (usually written as d(CGG CGG) where the mirror repeat is not apparent) can form other non-B-DNA structures, such as Z-DNA [64,65]. The c-MYC gene sequence provides two examples. One MR (the two repeats are underlined) can form GQ and H-DNA (5′-TGGGGAGGGTGGGGAGGGTGGGGAAGG-3′) while another MR (the two repeats are underlined) forms GQ or H-DNA (5′-GGGAGGGGCGCTTATGGGGAGGG-3′) [66,67,68].

In other cases, each strand from a duplex can individually fold to produce different non-canonical structures. For example, the d(GGGGCC) repeat from the C9ORF72 gene that predisposes to amyotrophic lateral sclerosis can fold as a GQ (Figure 3A). The complementary strand instead can form a TPX (Figure 3B, the third strand is drawn in fuchsia). This sequence is a MR when written as CCG GCC. Alternatively, four C-rich strands can form i-motifs (highlighted in green). Each pair of strands can also pair to form duplexes as they enter and leave the i-motif region. These duplexes can also adopt an alternative structure by forming Z-DNA at both ends (Figure 3C). Besides adding a third strand to form a TPX, a pair of single-stranded d(AT) rich sequences can base pair to create helices with parallel strands instead of the anti-parallel alignment found in canonical Watson–Crick B-DNA (Figure 3D). It is currently unknown if and when such complex structures fold in vivo. However, these examples reveal the propensity of some sequences to fold in multiple ways, including those that can form non-canonical triplets. Both the folds and the outcomes then depend on the context.

## 6. RNA Only TPXs and Other RNA Folds

RNA-only TPXs are also known and have recently been reviewed [69]. In cells, their role in RNA processing pathways varies. A TPX forms the catalytic core of the human U2–U6 spliceosome. A TPX fold also stabilizes the telomerase enzyme but is not part of the catalytic core [70,71] (Figure 3E). The TPX in MALAT1 RNA protects the 3′ end from nucleases [69]. Formation of the MALAT1 TPX depends on cleavage by RNase Z, a reaction that yields the mascRNA (Figure 3F with structures shown in 3G) [72]. A similar process is involved in the processing of the long NEAT1 pre-lncRNA. The tRNA-like molecules produced regulate inflammatory responses [73]. The MALAT1 TPX also binds METTL16 (methyltransferase 16, RNA N6-adenosine) through a protein-mediated VCR domain clamp. The bound METL16 then methylates adenosine 43 in the U6 snRNP, which is involved in splicing. How the different outcomes are related is not currently known [74,75,76].

## 7. Resolution of TPX

The use of RNA structural folds to scaffold the assembly of protein machines is quite widespread, as is well known from studies of ribosome assembly [77]. As with flipons, the RNA motifs rather than their sequence are critical to the assemblies created. Structural motifs are also important to their disassembly. The DExD/H-box RNA (DDX) helicases illustrate this principle [78]. The proteins initially undergo a low-affinity interaction with dsRNA. The deep grooves preclude the readout of their sequence. Instead, the helicases exploit structural features found in partially unwound dsRNA for targeting their interaction, with sequence-specific contacts are rarely made [79]. They instead make specific contacts with the phosphate backbone and the 2’OH group of the ribose sugar. The docking induces a conformational change that unwinds the RNA, resulting in a high-affinity interaction that reorients and separates the chains [79,80]. 

Paradoxically, many helicases are associated with particular sequence motifs [81]. This outcome may occur for several reasons. The helicase interactions may reflect the recognition of structural elements within a partially single-stranded RNA, such as K-turns [82,83], reverse K-turns [84], tetraloops and T-loops [85], and Z-turns [86] that nucleate RNA folding by bending and kinking the RNA backbone to create a distinct three-dimensional profile. The combination of distinct structural motifs bound by different helicase domains then codes for the interaction. Helicases may also bind to the partially single-stranded regions formed when flipons adopt an alternative conformation. These ssRNA sites are present at the junctions between a right-handed duplex and an alternative structure or in bulged regions or hairpins created to accommodate the flipon fold. The recognition by helicases of structural variations associated with G-flipons have been recently reviewed [36,87]. The sequence-specificity may arise for another reason. Helicases have different conformational states, with the transition from a closed to an open binding state triggered by another protein, often through a G-patch [88]. In this situation, the binding of an associated protein to a particular sequence motif can activate the helicase. These interactions with different sequence-specific proteins allow helicases to both enhance and suppress gene expression by dynamically remodeling the scaffolds bound. 

## 8. Genome-Wide Prediction of TPXs by Computational Approaches

Following the identification and characterization of TPXs in vitro, interest turned to their in vivo relevance. Numerous studies have been facilitated by computational approaches designed to predict TPX-forming sequences in genomes. These tools have recently been reviewed [89]. Algorithms like those used in Triplexator, Triplex Domain Finder, and an updated PATO version use canonical Hoogsteen pairing rules while routinely limiting predictions to sequences that are 16 bases or longer [90,91,92]. The programs detect TPX sequences enriched in DNA simple repeats. They find fewer TPXs in either low-complexity or imperfect repeats. Later methods, such as 3 plex, predict triplet formation for third strands as short as 8 nts and incorporates noncanonical triplets. The approach also considers the alternative non-TPX RNA folds that a sequence can adopt [93]. The LongTarget method instead prioritizes TPXs of 50 bases in length while tolerating a 30% error rate, including those mispairings from non-canonical triplets. The approach is optimized for detecting TPXs formed by long noncoding RNAs (lncRNAs) and uses various experimentally derived rules to score triplets [94]. The authors note that many predicted long TPXs are composites of shorter TPXs separated by non-triplet forming residues. In this scenario, a long TPX may have shorter segments with an sRNA as the third strand. This allows triplets to arise in a combinatorial manner, analogous to how protein transcription factors act together to regulate gene expression [93]. TripletAligner adopted a different strategy to train an Expectation-Maximization model using experimental data from triplexRNA-seq and triplexDNA-seq data from HeLa cells, with validation based on RADICL-seq and RNA ends on DNA capture mappings. The experimental substantiation of a limited number of predictions was also performed. The approach confirmed the accuracy of short 7–8 nt motifs enriched in the predictions.

As noted in a recent review, the computational tools vary considerably in their output, making the comparison of the results challenging [89]. Even with Triplexator, the search results for a TPX-forming Oligo (TFO) and a Triplex Target Site (TTS) in a promoter vary depending on the parameter used. For example, predictions are significantly impacted by simultaneously decreasing the TPX length from 19 to 15 nts and increasing the allowed mispairing rate from 10% to 20%, while minimizing the effects of noncanonical triplets by raising the required G-content from 25% to 65%. When a set of 27,876 sequences 1000 bp upstream and 200 bp downstream of a transcription start site is analyzed with the adjusted parameters, the number of predicted TPXs increases from 2596 to 6779 [90]. Interestingly, chromatin-associated RNAs were enriched for TTS, especially in GT-rich DNA sequences. As with any computational approach to a new problem, the initial TPX-focused methods captured classical motifs, while later methods detected other variants as the understanding of TPX-forming rules evolves.

## 9. Physical Mapping of TPXs In Vivo

Genome wide experimental studies to identify TPXs have also been developed. The chromatin-immunoprecipitation sequencing (ChIP-seq) approaches are currently limited due to a lack of TPX-specific antibodies. There is no antibody equivalent to the Z22 monoclonal antibody that allows for the detection of Z-DNA and Z-RNA, regardless of sequence [68], nor to the BG4 engineered antibody that binds to parallel-stranded G-quadruplex structures [95,96]. The three TPX reagents currently reported consist of an antiserum raised against d(A:T)*T [97,98] and the monoclonal antibodies JEL318 [99] and JEL466 [100] raised against d(T^m5^C:GA)*d(^m5^C^+^T)]. All three reagents have different, almost non-overlapping sequence-specificities [100], which is rather surprising given that the bases in TPXs are buried within the TPX. If the differences are due to heterogeneity in the backbones of each TPX type, structural studies may add clarity. Another possibility is that the antibodies recognize junctions or different features in the polymers used for validation.

Other experimental approaches for TPX detections are based on the copurification of chromatin-associated RNAs and DNAs. Protocols involve the proximity ligation of DNA to RNA, followed by selective sequencing of the covalently linked DNAs and RNAs. The method maps interactions between sequences that arise from non-contiguous chromosomal loci. DNase I, RNase A, and RNase H incubation steps are used in the protocols to increase specificity as TPXs are more resistant to these enzymes than single-stranded or double-stranded nucleic acids. Proteases allow the isolation of complexes stabilized primarily by nucleic acid interactions. The methods have been adapted to study particular protein complexes that can be enriched with appropriate antibodies. Genome-wide maps have been prepared using other techniques [101,102]. An alternative approach is based on the sequencing of the RNA that is co-immunoprecipitated with DNA (DNA-ip) combined with the sequencing of DNA pulled down by chromatin-bound RNA ligated to biotin-tagged linkers (RNA-capture). Computational methods can then help identify likely TTS [101]. The strengths and weaknesses of each approach have recently been reviewed [25]. The proximity ligation methods often produce short 22–28 nt sequences that do not contain the TTS and may be difficult to map if they involve TPX formation by repetitive elements. The sequences identified by DNA-ip and RNA-capture approaches have a 60% overlap, with the RNA-based isolation approach showing less specificity, with 20% of peaks found in annotated R-loop regions [101]. 

A recently published chemoproteomic approach is based on the intercalation of benzoquinoquinoxaline (BQQ) into TPXs [103]. The binding constants determined by Electrospray Ionization Mass Spectrometry for this compound in 1:1 and 1:2 complexes with TPX DNA were measured at K_1_ = 1.2 × 10^5^ M^−1^, K_2_ = 3.9 × 10^4^ M^−1^, respectively. In contrast, the values for the 1:1 and 2:1 complex for the DNA duplex were K_1_ = 3.2 × 10^4^ M^−1^ and K_2_ = 3.7 × 10^4^ M^−1^, respectively, offering at best a 25-fold increase in specificity for TPX detection [104]. In the cell-based approach, a derivative of BQQ was prepared by adding a photoreactive cross-linking reagent for proximity-dependent labeling and a linker for the biotin-mediated pull-down of associated proteins. Of the 78 proteins found in two cell lines, 13 were helicases. There were 11 proteins previously published as TPX binding proteins and 16 others that showed a preference (based on published datasets) for polyT sequences, with others showing a preference for mixed polypyrimidine motifs. Band-shift assays showed the specificity of the helicase DDX3X for triplets over duplexes and GQ [103].

These approaches can also be combined with other studies that utilize proximity ligation to identify DNA and RNA sequences or RNA sequences within 25 Å of each other. The motivation for these studies was to characterize chromatin-associated lncRNAs, their interactions, their relationship to RNAs transcribed at enhancers (eRNAs), and their sequence characteristics in an unbiased manner [102,105,106,107]. The analysis allowed the mapping of contacts within and between chromosomes, providing a high-resolution map of chromatin domains and nuclear structure. The overlay of ENCODE and other data for epigenetic marks, transcription binding sites, open chromatin regions, and flipon maps has provided new insights into non-canonical ways of encoding genetic information.

## 10. TPXs Forming Sequences and NFRs

Using these approaches, it is possible to ask how TPX formation influences the binding of nucleosomes to DNA. The analysis reveals that TTS are enriched in NFRs as they are not easily incorporated into nucleosomes [108,109]. Instead, TPXs enable the exact positioning of nucleosomes on DNA to create an NFR. This outcome is especially true for TPXs formed by adenosine-rich sequences (ARS). Even without forming TPX, ARS resist nucleosome formation. The duplexes are inherently rigid and bent due to the twisting of adenosine bases arising from the bifurcated hydrogen bonds formed in the major groove. These tensegrity struts formed resist deformation by bracing the successive layers of base pairs. Genome-wide, these features distorts the wrapping of nucleosomes around ARS, sometimes occluding binding sites for transcriptional factors [110]. The nucleosomes bound to ARS dissociate from DNA easily, unlike the stable nucleosomes present at −1 and +1 regions of an NFR [111]. They are enriched in NFRs and in promoters and play a role in phasing the downstream nucleosomes. However, DNA sequence alone cannot explain these outcomes in cells, as the recipient organism determines the phasing of a foreign DNA [112]. Further, recent time-course experiments suggest that the apparent enrichment of d(A:T) in fragile sites arises from the difference in sequence sensitivity to the nucleases used, rather than for other reasons [113].

ARS may also impact the positioning of nucleosomes involves by chromatin remodelers [114]. These sequences can render the movement of chromatin remodelers along DNA more energetically favorable in one direction, rather than the other [115]. In the yeast *Saccharomyces cerevisiae*, the ATP-dependent Remodeling the Structure of Chromatin (RSC) complex contributes to chromatin opening by preferentially displacing nucleosomes towards the 5′ end of a poly(dA:dT) tract and enhancing their disassembly [116,117]. Another complex, Imitation Switch 1a (ISW1a), which shrinks the NFR, preferentially moved nucleosomes in the opposite direction. A more elaborate system of periodic d(A_n_/T_n_) clusters (PATCs) is believed to function similarly in *Caenorhabditis elegans*. PATC boundaries prevent the silencing of essential early development genes by blocking the transitive spread of transposon suppression via RNA interference (RNAi). 

In mammalian genomes, the many repetitive sequences present also play a role in nucleosome phasing and epigenetic modification. SINEs localize the nucleosome over their A-rich regions and set the phase of adjacent nucleosome. Experimentally, these ARS can form TRX [101]. LINE-1s are also capable of TPX formation to create NFRs [52].

## 11. miRNAs and TPXs

Within cells, various factors can stabilize or promote TPX formation. Polyamines like spermidine and spermine counter the backbone charge and promote TPX formation at a neutral pH [118]. Other cationic proteins with suitably spaced charges can exhibit a similar behavior. Notably, TPX formation can be stabilized by nucleosomes through interactions with the N-terminal basic tail of Histone H3 (H3) based on in vitro assays, but not Histone H4 [119]. The in vivo phasing of nucleosomes was mapped by micrococcal nuclease sequencing in human HeLa cervical carcinoma cells and overlapped with Triplexator predictions [120]. TTS peaked at 150 nt on either side of the midpoint of nucleosomal contact with DNA. The TTS also overlapped with predicted microRNA seed sequence matches [121] and with H3K4me3 (H3 with lysine 4 trimethylated) marks associated with active promoters and H2K27ac (H3 with lysine 27 acetylated) flags for active enhancers. Modeling shows that the increased negative charge density of a polypyrimidine D:D*R TPX markedly increased the ionic bonds formed with histone H3 arginines and lysines, stabilizing the TPX. These included Arg2 and Arg8 residues that undergo unique epigenetic modifications [122,123]. The data reveal that TPX formation can modulate epigenetic marks, which in turn modulate TPX formation.

Besides miRNA, other sRNAs can have high tissue-specific expression, potentially impacting gene expression. Some are present in the nucleus, while others, such as tRNAs, shuttle between the cytoplasm and nucleus [124]. Due to their size, sRNAs only have a single TPX-forming site. The sRNA may be produced by the cleavage of longer RNAs, facilitated by nucleases such as Drosha, Dicer, and Argonaute and Piwi proteins, RNase Z during tRNA processing, integrator complexes that cause premature RNA transcriptional termination, splicing complexes that remove introns, and the polyA machinery that cleaves and tails transcripts [17]. Consequently, many sRNAs arise from the post-transcriptional processing of various transcript classes, including exons, introns, rRNAs, tRNAs, snRNAs, and snoRNAs [12]. Of these sRNAs, the evidence that miRNAs form TPXs is strongest for those that lack a mixed purine and pyrimidine in their sequence [121,125]. Bioinformatic analysis suggests that many miRNAs originated from and coevolved with SINEs, LINEs and other ERE [126,127]. Both piRNA and tRNA fragments also play an important role in regulating these elements by targeting argonaute family member effectors to these sequences [128,129,130]. As noted above, SINEs [101,120], and LINE-1 [52]. are also capable of TPX formation, suggesting that the interactions between EREs are coaptive.

## 12. lncRNAs and TPXs

While the sRNAs involved in TPX formation are produced in trans from distant sites, lncRNAs produced in cis (i.e., within the same chromosomal domain) also induce TPX formation to regulate gene expression. The lncRNAs can also act by controlling promoter access through the placement of nucleosomes. One example is the regulation by ribosomal RNA (rRNA) expression in mice by the sense pRNA (promoter RNA) and the antisense PAPAS (promoter and pre-rRNA antisense), neither of which is well conserved phylogenetically [131,132,133,134] (Figure 2B,C). The lncRNAs suppress rRNA transcription in two different ways, both involving TPX formation. The pRNA TPX inhibits binding by the enhancer protein TTF-1 (transcription termination factor 1) but instead preferentially binds DNMT3b (DNA methyltransferase 3 beta) that methylates a cytosine in the enhancer T_0_ at position 133 to decrease gene expression [133] (Figure 2B). The PAPAS TPX repositions nucleosomes by 24 nts to occlude the rRNA promoter [134]. It is unknown whether the lncRNAs remain bound to the chromosome after transcription, detach and form a bridge between the two chromosomal loci, or are processed to generate shorter TPX-forming fragments. Other examples of TPX-based promoter switches are known. For example, the *SPHK1* (sphingosine kinase 1) downstream promoter produces an antisense *KHPS1* lncRNA that initiates TPX formation at an upstream promoter, leading to enhanced expression of the sense transcript [135]. The human β-globin locus also employs a TPX switch to downregulate ε- and γ-globin upstream promoters and to enhance expression of the downstream β-globin TPX-forming transcript [136].

The lncRNAs involved with gene suppression generally have low expression and are tightly associated with chromatin [10]. The transcripts almost always arise from elements identified by the ENCODE project as enhancers, with bidirectional transcription common [137,138]. They may be intergenic, overlap protein-coding genes, or only introns. Some are spliced, capped, and polyadenylated, while others are not. Identifying these lncRNAs and mapping their interactions is challenging experimentally. The binding of lncRNAs with neighboring regions is most often reported [102], leading to models in which lncRNAs bring regions within the same chromosome together to promote epigenetic modifications or topological domain formation [135,136]. The bridging by lncRNAs of nearby enhancers to promoters may switch gene expression either on or off or to vary the RNA isoform produced [101,136,139].

Other lncRNAs, such as MALAT1 and NEAT1, are highly expressed and interact in sequence-specific ways with loci on other chromosomes, and can scaffold protein assemblies in both the nucleus and the cytoplasm [11]. The loss of a lncRNA can cause disease. In the case of phenylketonuria, the enzyme PAH (phenylalanine hydroxylase) activity is decreased by loss of the lncRNA *HULC* (hepatocellular carcinoma up-regulated long noncoding RNA) without any mutation of *PAH*. *HULC* facilitates PAH-substrate and PAH-cofactor interactions. In contrast, another intronic lncRNA *Pair* (PAH activating lincRNA) inhibits PAH activity [140]. LncRNAs from both mice and humans have also been associated with other visible phenotypes [11,141].

## 13. TPXs and EREs

The phasing of nucleosomes is also of importance in regulating expression of AEs, initially providing a means to halt transcription and retrotransposition of these SINEs [142]. Consequently, of the more than ~1.3 million AEs in the human genome, only 17,249 are actively transcribed by RNA polymerase 3 (Pol3), despite the intact A and B promoter boxes within these sequences able to engage Pol3 [143,144]. The few AEs transcribed by Pol3 exhibit cell-specific expressions from upstream cell-specific enhancer regions. Their transcription increase the readout of genes by RNA polymerase 2 (Pol2), with the AEs acting as enhancer SINEs (eSINEs) and likely leads to Z-DNA and GQ formation in the NFRs created [38,145]. Treatment with retinoic acid also induces the expression of Pol3 eSINEs through the DR2 repeats present in many of these elements, resulting in the enhanced expression of neighboring Pol2-transcribed genes. The small repeat-induced RNAs (riRNAs) produced from the AEs play a role in negatively regulating the cytoplasmic levels of Pol2 transcripts through Ago3-mediated RNA interference pathways [146].

In general, AEs within mRNAs are not well targeted by miRNAs due to their secondary structure and the sequence changes that result from RNA editing [147]. Yet studies identify miRNAs that map predominantly to the AE sense strand [127,148]. The possibility that these miRNAs act in the nucleus to phase nucleosomes over Alu elements has not been addressed, although the enrichment of TPX formation in Alu has been demonstrated in experimental studies [101,120]. However, enhancer regions are enriched for AEs that are also engaged by AGO1 protein bound to RNA [107,149]. AGO1 knockdown cell disrupts chromatin organization and globally alters gene expression [150,151]. Interestingly, the KCNQ1OT1 lncRNA induces genome-wide silencing of AEs through TPX formation involving the ARS found in AEs by localizing HP1α (encoded by *CBX5*) to the site [152].

Another notable example relates to the murine EREs that play an important role in bootstrapping the early steps of embryogenesis by activating the expression of zygotic genes (Figure 4) [153,154,155]. Expression of LINEs pre-implantation is associated with TPX formation, as detected with antiserum raised against d(A:T)*T (Figure 4C, adapted from the study of [153]). The later failure to repress LINE-1 transcription leads to developmental arrest [155]. The MTA (Mouse Transcript A) LTR drives the expression of embryogenic genes at the two-cell stage and is necessary to initiate the transition from maternal programming to embryonic gene activation [156].

In this pathway, the MTA promoter drives the production of a chimeric transcript by splicing the retroviral RNA to the transcript from the downstream gene. The MTA acts as an alternative promoter and encodes the first exon for a subset of genes [156]. Later, the MTA transcript undergoes cleavage rather than splicing, with the downstream gene promoter then driving gene transcription. The MTA element contains adjacent ERE splice and polyadenylation motifs (Figure 4A,B,H). The nucleosome placement may either favor splicing, or instead support transcript cleavage at the AAUAA site. Transcription from the MTA promoter likely favors the former, while antisense transcription from the downstream promoter would favor the latter. In both cases, chromatin remodelers that move nucleosomes in the direction of transcription finalize the placement of nucleosomes. The outcome depends on which promoter is currently active.

The MTA sequences encodes other regulatory elements that likely effect the switch in promoter usage. These include TPX-forming sequences predicted by the PATO program [92] (Figure 4C,D). A potential long TPX-forming sequence lies between positions 328 and 353 of the MTA sequence and just 3′ to the AAUAA cleavage site (Figure 4G,H). TPX formation could result from the fold back of the spliced transcript initiated from the MTA promoter. The TPX formed would facilitate use of the downstream cellular promoter by positioning nucleosomes to block splicing and promoting cleavage of the MTA transcript. The upstream transcript would be lost. The downstream promoter then is no longer subject to transcriptional interference mediated by processing of this RNA. The outcome is similar to the regulation of the rRNA gene by *PAPAS* (Figure 2). A positive feedback loop could then develop where the piR-4482 cleaved from the antisense transcript precludes the future use of the MTA promoter by directing its incorporation into constitutive heterochromatin (Figure 4F,G,H). The antisense transcript would also enable recruitment of tissue-specific enhancers to further amplify transcription from the cellular promoter.

## 14. lncRNAs, sRNAs, and Chromatin Condensates

The interplay between lncRNAs and sRNAs with promoter transcripts enables various regulatory interactions, including both cis and trans interactions, beyond the creation of NFRs [157]. The NFRs created with D:D*R specify the available transcription factor binding sites and flipons in the promoter and enhancer regions. The sense and antisense transcripts produced can then impact the expression of genes in the neighborhood by transcriptional interference, as described for the lncRNA upstream of the dihydrofolate reductase gene [158]. It was proposed in this case that TPX formation prevents TFIIB engagement with DNA. Other lncRNAs acting in cis also produce transcriptional interference, although TPX formation has not been experimentally assessed. The lncRNAs include *Chaserr* (*CHD2*-adjacent, suppressive regulatory RNA), which downregulates *Chd22* (Chromodomain Helicase DNA Binding Protein 2) [159], and the *Airn*-induced H4K9me3 of the *Igfr2* gene. In both cases, the only requirement was the overlap of the lncRNA transcript with the gene promoter [160].

A subset of lncRNAs recruits the polycomb repressive complex 2 (PRC2) to promoters through TPX formation, with the PRC2 enhancer of zeste 2 polycomb repressive complex 2 subunit (EZH2) docking to a GQ formed by the lncRNA [161]. Examples of such lncRNAs include *MEG3*, which targets genes in the TGFβ (transforming growth factor beta) pathway [139], the heart and body wall-specific *Fendrr*, the radiation-induced *Particle*, and the WNT pathway regulator *HOTAIR* lncRNAs [162,163,164]. A computational analysis of super enhancers (SEs), defined as enhancers in close genomic proximity with unusually high levels of mediator binding, from 27 different human cell and tissue types identified 442 unique super-long noncoding RNA (super-lncRNA) transcripts, 70% of which were tissue-restricted and 80% of which had at least a single TPX-forming site. Triplexator predicted that, on average, each super-lncRNA can form D:D*R TPXs with 25 SEs, with a single SE targeted by multiple lncRNAs at many different, but evenly distributed locations (10–50 sites on average) [149]. Over 40% of SE sites were predicted to bind super-lncRNAs derived from EREs, with 34% from SINEs, 9% from LINEs, 4% from LTRs, and 2% from DNA transposons. A separate study revealed that the distribution of Alu elements within lncRNAs differed notably from coding genes, with Alus distributed throughout, with an increase in right arm monomers in the central region [165].

The outcomes also depend on the sequence features of the lncRNAs anchored through TPX formation (Figure 5A). Each lncRNAs delivers a set of protein-binding motifs that form a scaffold for assembling different cellular machines with various functionalities [166]. The lncRNAs can rapidly evolve by adding or subtracting sequence motifs to modify the condensates they nucleate [51]. They lack the sequence restraints that restrict the variation in protein-coding transcripts and represent a rapid way to evolve novel nuclear architectures [167]. The sequence overlap between different lncRNAs also provides functional redundancy. The shared TPX forming and protein binding motifs allows other lncRNAs to compensate for the loss of any one lncRNA.

Gene regulation may involve sRNAs processed from lncRNAs. For example, the H19 lncRNA, the first lncRNA identified, encodes miR-675. The gene is highly abundant in embryonic development and is important in skeletal muscle development [168,169,170]. The induction of gene silencing is possible with miRNAs that target sites in chromatin-associated RNA many kilobases away from an active promoter [171]. The miRNAs themselves may be produced by nearby ncRNAs. Indeed, it is possible to identify a subset of 303 miRNAs that are nuclear, produced from chromosomal regions with active chromosomal marks, and whose expression correlates with nearby genes, known as nuclear activating miRNAs (namiRNAs) [172,173]. These miRNAs can be overlapped with a set of high-confidence miRNAs (hc-miRNA, n = 504) [174] and with a set of miRNAs, called saRNA, that are proposed to activate gene expression. The saRNA were identified using RNA ChIP seq with Pol 2 antibodies. They have sequence matches within 20 bases of a promoter TATA box and also have a minimum free energy less than −27.6 kcal/mole. There are 72 miRNAs in the overlap of hc-miRNA, namiRNA, and saRNA (listed in Figure 5C).

## 15. Helicases and Enhancer–Promoter RNA Interactions

Interestingly, in the case of miR-34a, an interaction between the AGO2-bound miRNA activates the RNA helicase DDX21. The enzyme then induces the release of Pol2 from the pause site through an interaction with the 7SK complex that otherwise stalls transcript elongation [175]. In other studies, an interaction between 7SK and the Brahma-associated factor (BAF) chromatin remodeler delineates the NFR of enhancers and is particularly important for demarcating the SE [176]. Phasing nucleosomes in SEs is necessary to counter the convergent transcription arising from such closely spaced transcriptional units, both in cells and during development. The precise localization of nucleosomes is particularly important during embryogenesis, when the depletion of parental chromatin leads to pervasive transcription [177]. The nucleosome placement involves an embryonic version of the BAF [178]. Interestingly, DDX21 is also involved in the interaction between 7SK and the BAF. The finding is consistent with a model where AGO-bound miRNAs localize the complex (Figure 5B).

The different interactions involving helicases may also explain some of the divergent results on transcription obtained with nuclear miRNAs (Figure 5B). Previously the different outcomes were thought to depend on whether or not the target-bound miRNA activated the AGO2 slicer activity and transcript cleavage [179]. Instead, the outcome may depend on whether the miRNA complex localizes DDX12 close enough to a paused polymerase to disassemble the 7SK ribonucleoprotein and initiate transcript elongation. In contrast, other interactions that place DDX12 near BAF may help phase of nucleosomes in enhancers, or activate the little elongation complex that drives snoRNA transcription by Pol2 [180] (Figure 5B). Interestingly, the regulation of Pol1 transcription by a snoRNA complex also depends upon DDX21 [181]. 

At promoters, the miRNA guided engagement of AGO may enable the resolution of TRP by DDX3X, GQ by DHX36, and dsRNA by DHX9. The complexes formed then decap prematurely terminated transcripts, resolve R-loops, and recycle the factors necessary to reestablish a pre-initiation complex [28,182,183,184]. The role of helicases in modulating flipon conformation has also been recently reviewed [38,185], as has the setting of flipon conformation by miRNAs [186]. In all these cases, helicases enable setting and resetting of gene expression. The helicases involved mostly differ from those involved in the canonical cytoplasmic miRNA pathway, where the interaction of AGO2 with DDX6 usually initiates the engagement of the microRNA-induced silencing complex [187]. Other AGO proteins with helicases are known, but the regulatory interactions are less well characterized [188,189]. 

ENCODE data capture many of the interactions at promoters between flipons and helicases (Figure 5D,E). The overlap of AGO, DDX3X, and experimentally determined triplex formation is shown for the SPHK1 gene. There are no R-loops reported in this region [101], Both DDX3X and DHX36 localize over different regions of the insulin-like growth factor 1 receptor (IGFR1) gene. DDX3X and AGO2 overlap, while DHX36 is detected over R-loop regions where G-quadruplexes can form. Both of the gene promoters are in CpG-rich regions of the genome. They are both bound by the CCCTC-binding factor (CTCF) that facilitates the formation of chromatin loops.

## 16. Summary

The transactions between flipons and condensates influence gene expression by defining the NFR and forming RNA clouds that position protein complexes at enhancers and promoters (Figure 6). The RNA helicases color the chromatin in various hues through the interplay of lncRNAs with sRNAs and facilitate the transition from one chromatin state to another. Flipons play an important role in these processes. T-flipons set the stage by phasing nucleosomes, demarcating both the available transcription factor binding sites and the G- and Z-flipons necessary to initiate and reset promoters for the next round of transcription. The RNAs produced from these open regions amplify these effects by increasing the number of available protein binding sites to capture the proteins released when promoters are reset. This ensures the proteins are available to initiate the next round of transcription. This process also facilitates the removal of proteins from DNA during the transition between active and suppressed states. At each stage, the RNA cloud also turns over. The churn is energetically costly, but facilitates promoter updates based on the current cellular state. The cycles created and counteracted constantly recolor chromatin to channel cell fate.

## 17. Future

The current view of the genome and the role of repeat sequences is undergoing rapid revision. New technologies and concepts are improving our understanding of how cells evolve and function. These insights will lead to novel strategies for treating disease by resetting disease states and preventing their progression. The targeting of TPXs is now aided by the ENCODE mapping of cell-specific regulatory sites [8] and the advances in nucleic acid chemistries for clinical applications [190]. Vector-based delivery systems for circRNAs also promise new therapeutic approaches. None of these technologies were available when TFO therapeutics were first investigated at the turn of the century. Everything was trial and error, with no clinical candidates emerging [191].

Now with our high-resolution maps of RNA and DNA interactions and our newfound understanding of nuclear dynamics, we have fresh insights on how to proceed. One current challenge is to develop strategies to reposition nucleosomes and recolor cells’ epigenetic markings. This approach aims to change the enhancer-promoter pairs that drive gene expression. Therapeutics based on circRNAs are an interesting area to explore, given the increased stability and specificity of the TPX formed. Another strategy is to target the condensates as they recycle. Therapeutics that target helicases to a particular location in the RNA cloud could potentially enhance or suppress gene expression. Therapeutics that bind loops or bulges in DNA produced when flipons adopt different conformations may produce the same outcome by changing the active flipon in a particular NFR. The proposed strategies will benefit from recently developed computational tools to map RNA interaction motifs in chromatin [192]. All these approaches are sequence-specific and allow for prolonged dosing of the therapeutic, as already demonstrated for anti-sense and siRNA therapeutics. Given that the delivery of nucleic acid therapeutics to cells via gymnosis is possible, but only in limited amounts, therapeutics that activate existing positive feedback loops in a cell are more likely to see success in the clinic [193]. 

## Figures and Tables

**Figure 1 ijms-26-04032-f001:**
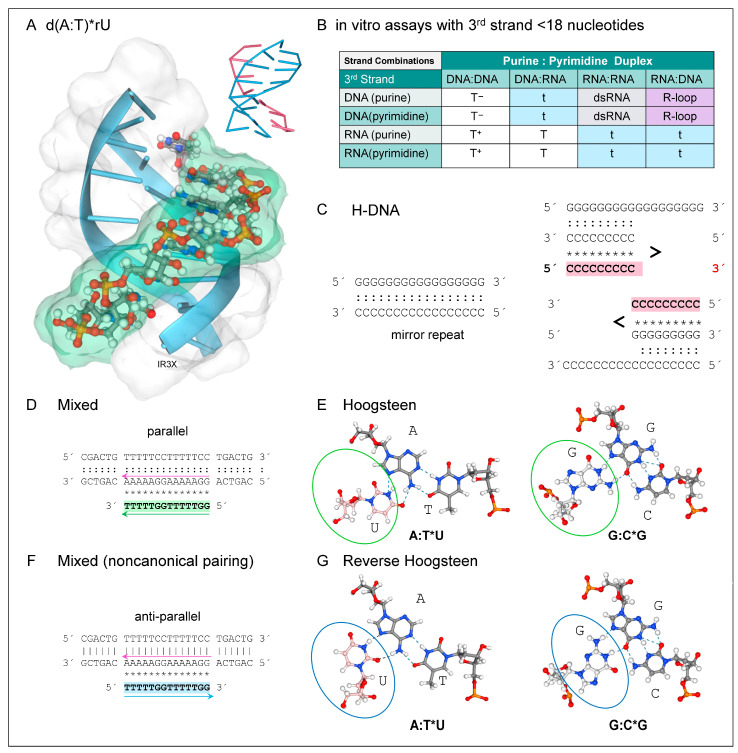
In a TPX, the orientation of the third stand is antiparallel to that of the matching pyrimidine or purine-rich strand within the double helix. To avoid confusion, TPXs are named with reference to the 5′ to 3′ polarity of the purine strand: polypyrimidine TPXs are called parallel, and polypurine TPXs are named anti-parallel TPXs. In the text and figures, Watson-Crick hydrogen bonds in the duplex are shown as “:” and Hoogsteen bonds formed with the third strand are denoted by “*”. DNA helical strands are designated by “d” and RNA strands by “r” (**A**) The canonical d(A:T)_n_*rU TPX with the third strand shown as a ball and stick model in a green-tinted surface representation. The inset is of a wireframe model for the TPX viewed from behind. (**B**) The in vitro stability of TPXs depends on the ordering of the RNA and DNA strands. A helix with an RNA purine strand is not observed to form a TPX. Rather, the R:D helix is more energetically favored. (**C**) The d(G)_n_ mirror repeat sequence can form a TPX with a purine strand. Under acid conditions, when cytosine is protonated, a TPX with a pyrimidine strand forms instead. (**D**) TPXs can dock a third strand composed of G and T bases in a parallel. (**E**) The parallel “mixed” TPXs form with Hoogsteen hydrogen bonds (**F**) The alternative anti-parallel orientation formed with the same sequences. (**G**) Here, reverse-Hoogsteen hydrogen bonds form between the duplex and the third strand. In panels (**C**,**D**,**F**), the third strand of the TPX has a colored background.

**Figure 2 ijms-26-04032-f002:**
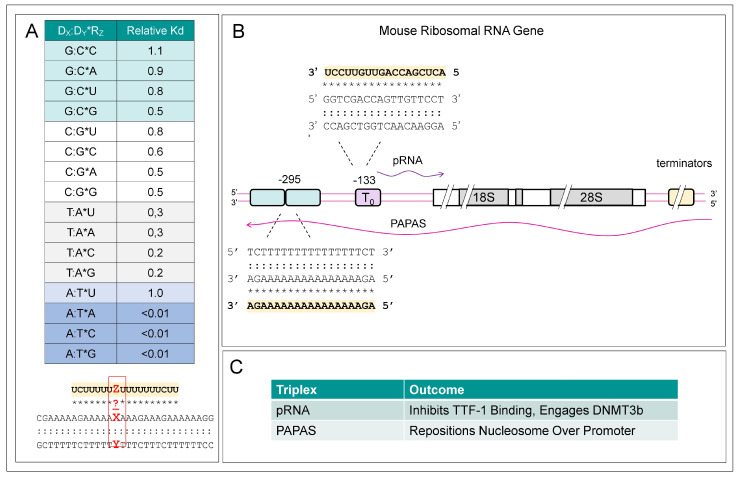
Noncanonical triplets and ribosomal gene regulation: (**A**) Substitution of a central triplet in a d(T:A)*U TPX (boxed in red) with a noncanonical triplet allows an estimate of their relative stability (adapted from [62]). Watson-Crick basepairs are indicated by “:” and Hoogsteen basepairs by “*”. (**B**) In mice, but not yet shown in humans, the sense pRNA lncRNA and the antisense PAPAS lncRNA form TPXs that suppress rRNA transcription, as explained in the text. (**C**) While pRNA induces DNA methylation by DNMT3b of the enhancer T_0_, PAPAS repositions a nucleosome to occlude the promoter. The third strand of each TPX has a colored background.

**Figure 3 ijms-26-04032-f003:**
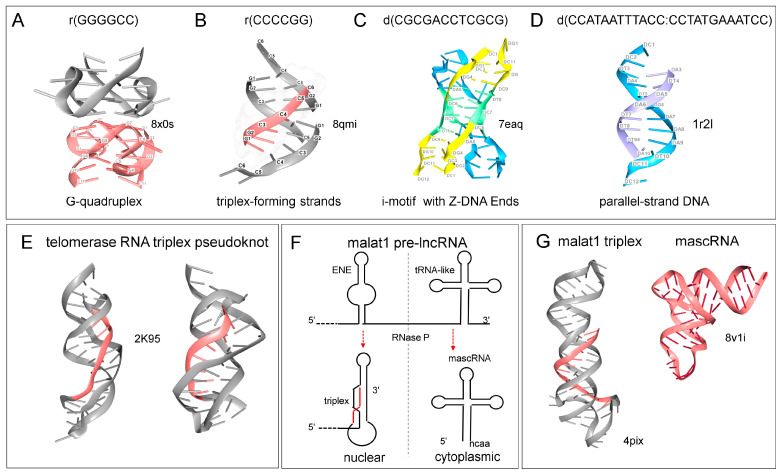
The diversity of folds formed by simple repeats: (**A**) One strand of the C9ORF72 repeat can fold with another segment to form a pair of GQs that stack on each other. (**B**) In contrast, the complementary strand can fold to form a TPX as the sequence CCGGCC is a MR. The third strand is colored fuchsia. (**C**) Four C-rich strands can form an i-motif (colored green) that can adopt complex folds, including one where each pair of strands (one pair colored blue and the other yellow) forms Z-DNA at both ends. (**D**) Two A-rich sequences can form a parallel-strand right-handed helix rather than folding with a third strand to form a TPX. (**E**) The telomerase pseudoknot also contains a TPX. Views from the front and back are shown. The third strand of the telomerase TPX is colored fuchsia. (**F**) The processing of the MALATI pre-lncRNA that produces a TPX at the end of MALAT1 and the tRNA-like mascRNA. The third strand of the MALAT TPX is colored fuchsia, (**G**) The structures of the MALAT1 TPX, tRNA-like mascRNA. The third strand of the MALAT TPX is colored fuchsia. Protein database (PDB) accession codes for the structures are given in each panel.

**Figure 4 ijms-26-04032-f004:**
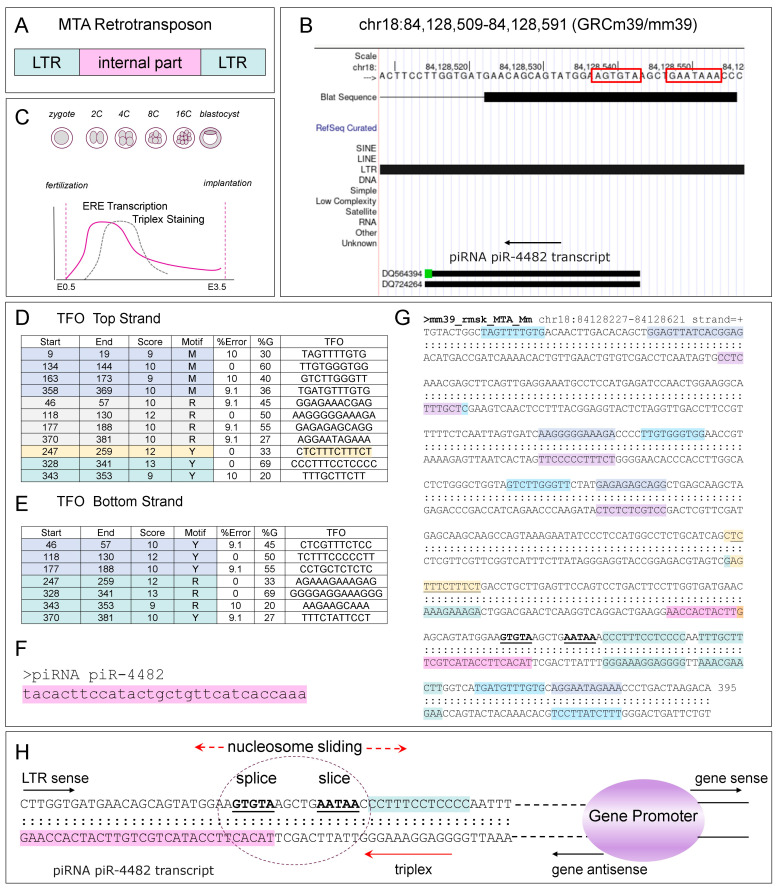
MTA Long Terminal Repeat (LTR) retrotransposon plays important roles in early embryonic development. (**A**) The structure of the MTA LTR. (**B**) The 369 nt MTA transcript shown contains a splice donor site AGTGT and a AAUA cleavage site close to each other (both boxed in red). The antisense strand encodes the piR-4482 piRNA. (**C**) Like other EREs at the 2-cell stage, MTAs are highly expressed. Slightly later, TPX formation detected with a d(A:T)*T antiserum also peaks (adapted from the study of [153]). (**D**,**E**) show sequences and positions of TPX-forming sequences in the MTA sequence on the top and bottom strands. They are color coded by whether they have a pyrimidine (Y), purine (R), or mixed (M) purine and pyrimidine third strand. (**F**) The sequence of piR-4482 (fuchsia background color). (**G**) The sequence context of the potential TPX-forming sequences (using the same color coding as in (**D**,**E**), relative to the splice and cleavage sites (in bold font and underlined). (**H**) A dotted circle shows the location of the splice and cleavage sites in the MTA promoter region. The piR-4482 sequence (with a fuchsia background color) overlaps the splice site. A long TPX-forming sequence (position 328 to 353) is immediately 3′ to the cleavage site that are underlined and in bold font. These two sequence elements likely impact the switch from the MTA throughout the genome to the cellular promoters essential for bootstrapping zygotic gene expression.

**Figure 5 ijms-26-04032-f005:**
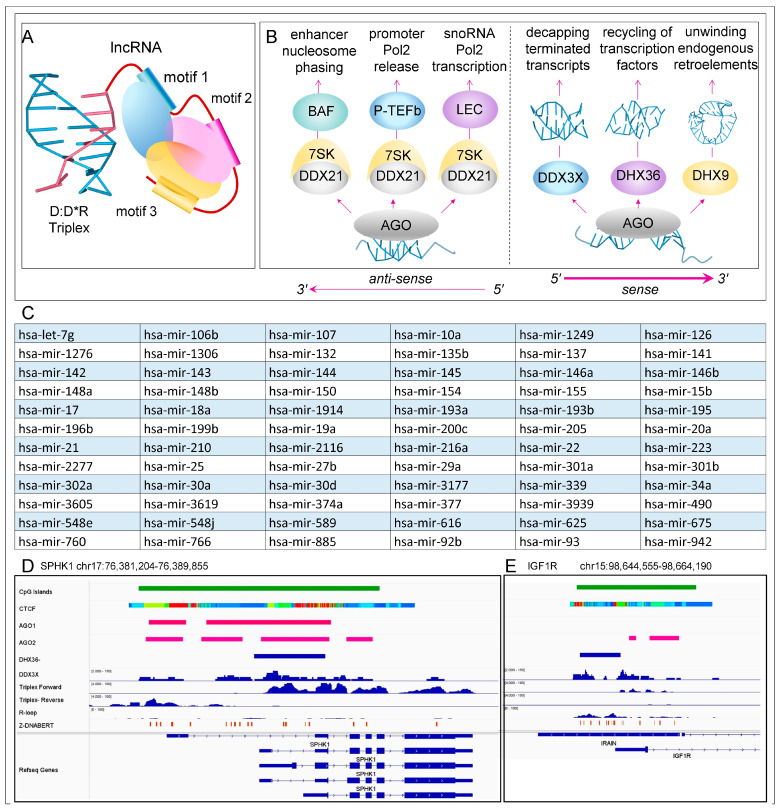
Noncoding RNA effectors: (**A**) lncRNAs anchored by TPX formation can localize different protein machines to a site through the motifs they contain. (**B**) Through various interactions with RNA helicases, AGO-bound miRNAs can activate Pol2 transcription. At enhancers, DDX2 acts through the Brahma-associated factor (BAF) complex to phase nucleosomes. At promoters, the DDX21 helicase disassembles the 7SK ribonucleoprotein to release Pol 2 from the pause site of cellular genes and also stimulates the transcription of snoRNAs through the little elongation complex (LEC). A different set of helicases (DDX3X, DHX9, and DHX36) enables the resolution of different flipon structures formed by RNA to release bound proteins and that decap prematurely terminated transcripts, enabling their removal by nucleases. (**C**) miRNAs that potentially activate transcription, derived by overlapping sets of well-validated miRNAs that are encoded by enhancers, bound by Pol2, and bind sequences within 20 nts of a promoter TATA box at minimum free energies of less than −27.6 kcal/mole. (**D**,**E**) Examples from the human *SPHK1* and *IGFR1* genes showing the overlap of helicase and argonaute protein binding with experimentally determined TPX using data from the ChIP-Atlas resource (https://chip-atlas.org/peak_browser (accessed on 8 April 2025)) and GSE120850 [101]. Genomic coordinates are from hg38.

**Figure 6 ijms-26-04032-f006:**
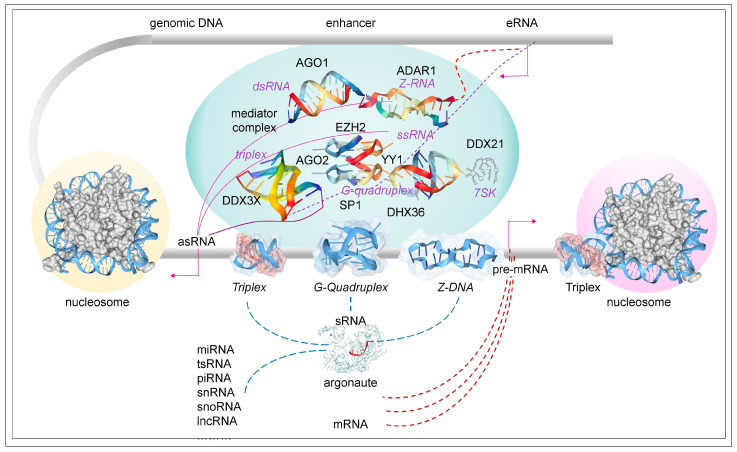
TPX Biology. Both lncRNAs and sRNAs play roles in the phasing of nucleosomes by chromatin remodelers. This process creates nucleosome-free regions that expose transcription binding sites and free flipons to adopt alternative conformations that are important for reinitiating transcription after each cycle [49]. The enhancer (red dashed lines) and antisense RNAs (purple solid lines) interact to form condensates that form various flipon structures and partially single-stranded regions to which helicases bind. The RNA cloud localizes chromatin-modifying complexes that color the histones with different epigenetic marks. Proteins that interact with flipons can partition between DNA and DNA binding sites to allow each to reset and to maintain each at a high local concentration. The turnover of condensates by helicases enables updates from inputs from elsewhere in the cell, including sRNAs, proteins, and signaling molecules. The processing of ncRNAs that generates argonaute bound sRNAs can occur in the nucleus (blue dashed lines) or in the cytoplasm (brown dashed lines). The different flipon RNA structures formed are tagged with purple, italicized labels. The arrows parallel to the genomic DNA show the direction of transcription. The placement of nucleosomes and flipons in enhancers are not shown but are similar to those drawn for promoters. Abbreviations: dsRNA, double-stranded RNA; ssRNA, single-stranded RNA; asRNA, antisense RNA; and sRNA, small RNA.

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
