# Peer review of "Triplexes Color the Chromaverse by Modulating Nucleosome Phasing and Anchoring Chromatin Condensates"

_ijms, 2025, doi:10.3390/ijms26094032_

Round 1
Reviewer 1 Report
Comments and Suggestions for Authors
In this manuscript, Alan Herbert reviews the current knowledge on DNA:RNA triplexes and their influence on chromatin composition, gene expression, gene splicing, phenotype, and disease appearance and progression. The topic is important for enhancing our knowledge of basic biological processes on the molecular level (transcription, translation, gene splicing, chromatin structure control, embryogenesis, and cellular programming) as well as for guiding the development of new drugs.
The manuscript comprehensively covers the topic, but I cannot yet recommend it for publication. The text is difficult to follow and suffers from occasional typos and editing errors.
My suggestions to the author are next:
- Reduce the usage of abbreviations and try to express the message with more accessible language. The aim of the manuscript is to offer a comprehensive review of the topic of triplex DNA/RNA structures and their role in gene regulation and cell health.
- Include additional tables and figures and present triplexes via their structure and role. The manuscript starts with the mechanisms of formation, then covers different roles, and ends with the role of RNA-only triplexes. Can you cover structures and roles with additional tables/figures?
- Use shorter sentences, they are easier to follow.
- Some sentences can be edited for clarity, e.g., the sentence on line 425: “The sense and antisense transcripts produced can then impact the expression of the genes in the neighborhood by transcriptional interference, as described for the lncRNA upstream of the dihydrofolate reductase gene [118].” can be edited as “The sense and antisense transcripts can impact the expression of the genes in the neighborhood by transcriptional interference, as described for the lncRNA upstream of the dihydrofolate reductase gene [118].”. The deletion of the words ‘produced’ and ‘then’ makes for a much smoother flow.
- There are some typos:
a. Line 193. There is a full stop in the middle of the sentence, or there are two sentences?
b. Line 263. The sentence is interrupted in the middle by a full stop. The whole sequence itself is difficult to follow.
6. First, define a term and then provide the abbreviation in parentheses, not the other way around.
7. Improve the figures, especially the depiction of triplexes. A subfigure can be added to Figure 1 to depict three strands using only cartoon representation (before or after Figure 1a).
The clarity of Figures 4D-H can be improved by using different colors for different strands.
8. When describing computational tools for the analysis of triplets, you may mention works by Schlick and collaborators on the graph representation of RNAs, for example:
-
-
- Qiyao Zhu, Louis Petingi, and Tamar Schlick, "RNA-As-Graphs Motif Atlas—Dual Graph Library of RNA Modules and Viral Frameshifting-Element Applications", Int. J. Mol. Sci., https://doi.org/10.3390/ijms23169249, 23:9249, 2022.
- Qiyao Zhu and Tamar Schlick, "A Fiedler Vector Scoring Approach for Novel RNA Motif Selection", Journal of Physical Chemistry B, doi: https://doi.org/10.1021/acs.jpcb.0c10685, 125(4):1144–1155 (2021).
- Stephanie Portillo, and Tamar Schlick, "Bridging chromatin structure and function over a range of experimental spatial and temporal scales by molecular modeling", WIREs Computational Molecular Science, 10(2): e1434 (2019).
- Ognjen Perisic, and Tamar Schlick, "Computational strategies to address chromatin structure problems", Phys Biol., 13(3): 035006 (2016).
-
The review articles contain well-organized tables and figures dealing with the problem of chromatin modeling."
- Check whether Alu elements are ERE. Endogenous retroelements (ERE)stem from retroviruses, while Alu elements “probably originated from a 7SL RNA gene after deletion of a central sequence”, see https://en.wikipedia.org/wiki/Signal_recognition_particle_RNA
While writing this review, my intention was not to hurt the feelings of the author. I can see that the author spent considerable time doing research and working on the text, and I know from my own experience how difficult science can be. It is better for the author to face the criticism of one reviewer than to face the criticism of a much broader audience.
I believe that if improved, this work can be a very important contribution to the study of DNA:RNA triplexes. The author has enough experience and knowledge to improve the manuscript. With additional work and time, this can become a very publishable review study and an excellent introduction to the field to a wider audience.
Comments on the Quality of English Language
- Reduce the usage of abbreviations and try to express the message with more accessible language. The aim of the manuscript is to offer a comprehensive review of the topic of triplex DNA/RNA structures and their role in gene regulation and cell health.
- Use shorter sentences, they are easier to follow.
- Some sentences can be edited for clarity, e.g., the sentence on line 425: “The sense and antisense transcripts produced can then impact the expression of the genes in the neighborhood by transcriptional interference, as described for the lncRNA upstream of the dihydrofolate reductase gene [118].” can be edited as “The sense and antisense transcripts can impact the expression of the genes in the neighborhood by transcriptional interference, as described for the lncRNA upstream of the dihydrofolate reductase gene [118].”. The deletion of the words ‘produced’ and ‘then’ makes for a much smoother flow.
- There are some typos:
a. Line 193. There is a full stop in the middle of the sentence, or there are two sentences?
b. Line 263. The sentence is interrupted in the middle by a full stop. The whole sequence itself is difficult to follow.
5. First, define a term and then provide the abbreviation in parentheses, not the other way around.
Author Response
Thanks for you helpful comments!
- Reduce the usage of abbreviations and try to express the message with more accessible language. The aim of the manuscript is to offer a comprehensive review of the topic of triplex DNA/RNA structures and their role in gene regulation and cell health.
I have better defined the abbreviations and provided expanded explanations so that the manuscript is easier to follow
- Include additional tables and figures and present triplexes via their structure and role. The manuscript starts with the mechanisms of formation, then covers different roles, and ends with the role of RNA-only triplexes. Can you cover structures and roles with additional tables/figures?
I have added two new figures and modified others to make them more explanatory and also to include experimental data.
- Use shorter sentences, they are easier to follow.
The manuscript has been extensively rewritten.
- Some sentences can be edited for clarity, e.g., the sentence on line 425: “The sense and antisense transcripts produced can then impact the expression of the genes in the neighborhood by transcriptional interference, as described for the lncRNA upstream of the dihydrofolate reductase gene [118].” can be edited as “The sense and antisense transcripts can impact the expression of the genes in the neighborhood by transcriptional interference, as described for the lncRNA upstream of the dihydrofolate reductase gene [118].”. Deleting the words ‘produced’ and ‘then’ makes for a much smoother flow.
I have used Grammarly to improve sentence structure
- There are some typos:
Thanks for pointing these out. These are now all corrected
- Line 193. There is a full stop in the middle of the sentence, or there are two sentences?
- Line 263. The sentence is interrupted in the middle by a full stop. The whole sequence itself is difficult to follow.
- First, define a term and then provide the abbreviation in parentheses, not the other way around.
I have done so.
- Improve the figures, especially the depiction of triplexes. A subfigure can be added to Figure 1 to depict three strands using only cartoon representation (before or after Figure 1a).
The clarity of Figures 4D-H can be improved by using different colors for different strands.
I added a sub-figure and two other figures
- When describing computational tools for the analysis of triplets, you may mention works by Schlick and collaborators on the graph representation of RNAs, for example:
- Qiyao Zhu, Louis Petingi, and Tamar Schlick, "RNA-As-Graphs Motif Atlas—Dual Graph Library of RNA Modules and Viral Frameshifting-Element Applications", Int. J. Mol. Sci., https://doi.org/10.3390/ijms23169249, 23:9249, 2022.
- Qiyao Zhu and Tamar Schlick, "A Fiedler Vector Scoring Approach for Novel RNA Motif Selection", Journal of Physical Chemistry B, doi: https://doi.org/10.1021/acs.jpcb.0c10685, 125(4):1144–1155 (2021).
- Stephanie Portillo, and Tamar Schlick, "Bridging chromatin structure and function over a range of experimental spatial and temporal scales by molecular modeling", WIREs Computational Molecular Science, 10(2): e1434 (2019).
- Ognjen Perisic, and Tamar Schlick, "Computational strategies to address chromatin structure problems", Phys Biol., 13(3): 035006 (2016).
I added reference c
The review articles contain well-organized tables and figures dealing with the problem of chromatin modeling."
- Check whether Alu elements are ERE. Endogenous retroelements (ERE)stem from retroviruses, while Alu elements “probably originated from a 7SL RNA gene after deletion of a central sequence”, see https://en.wikipedia.org/wiki/Signal_recognition_particle_RNA
Alu are nonautonomous retrotransposons
While writing this review, my intention was not to hurt the author's feelings. I can see that the author spent considerable time doing research and working on the text, and I know from my own experience how difficult science can be. It is better for the author to face the criticism of one reviewer than to face the criticism of a much broader audience.
No hurt at all
I believe that if improved, this work can be a very important contribution to the study of DNA:RNA triplexes. The author has enough experience and knowledge to improve the manuscript. With additional work and time, this can become a very publishable review study and an excellent introduction to the field to a wider audience.
I think the paper is much improved and synthesizes a lot of material in a novel manner. You are correct – it took a lot of additional time to assemble the pieces.
Reviewer 2 Report
Comments and Suggestions for Authors
The review paper titled “Triplexes Color the Chromaverse by Modulating Nucleosome Phasing and Chromatin Conformations” by Alan Herbert presents an interesting topic on triplex DNA structures (T-flipons) and their role in chromatin remodeling, gene regulation, and cellular programming. It covers multiple aspects, including triplex formation mechanisms, interactions with lncRNAs and circRNAs, and their regulatory roles. The discussion on therapeutic potential and emerging technologies adds relevance. In my opinion the scope fits well in the IJMS and it is clearly written and easy to follow. However, before this review paper is published the following minor comments needs to be addressed:
- Reference appear incorrectly formatted e.g. lane #435 {Latos, 2012 #3542]
- Lane 58: “Before these studies, roles for the noncoding part of the genome were not explored, or were considered unlikely." This is misleading; non-coding RNA research (e.g., lncRNAs, miRNAs) has been a major focus for decades.
- Lane 65: "At best, the persistence of such sequences in the genome was considered as one of those quirks of Nature’s imperfection." This is an opinionated statement that needs stronger evidence.
- Terminology including "flipons" and "chromaverse" is not widely recognized and should be better defined.
- Several sentences are grammatically unclear, and overgeneralized claims (e.g., about the historical understanding of non-coding regions) need more nuanced discussion and stronger citations.
- This review needs a strong conclusion section. In the last paragraph it mentions rapid revisions in genomic understanding and new technologies, however it does not clearly state what these specific revisions or technologies are, nor does it provide concrete examples of breakthroughs. Also, phrases like "these insights lead to novel strategies" and "many of these advances may be facilitated" remain abstract. What specific diseases or applications are being targeted? How do these strategies compare to existing ones?
- Also, the last section shifts focus to triplexes and therapeutic applications without clearly tying them back to flipons. If flipons are central to the review, their future research directions should be explicitly stated. A future outlook should acknowledge limitations or unanswered questions in the field, what are the biggest hurdles in applying these technologies?
Author Response
Thanks for your helpful feedback!
- Lane 58: “Before these studies, roles for the noncoding part of the genome were not explored, or were considered unlikely." This is misleading; non-coding RNA research (e.g., lncRNAs, miRNAs) has been a major focus for decades.
That’s funny – it was not my opinion but rather it came from Sidney Brenner, who was very opinionated. I have added the reference and more of the historical detail as this is likely not widely known by many outside the field.
- Lane 65: "At best, the persistence of such sequences in the genome was considered as one of those quirks of Nature’s imperfection." This is an opinionated statement that needs stronger evidence.
Again, historically accurate.
- Terminology including "flipons" and "chromaverse" is not widely recognized and should be better defined.
I have defined these more clearly. The term flipon has appeared in many different journals, including those from MDPI, the Royal Society, JBC, Nature.. I think there are more than 30 publications on flipons. I also described the background to the chromaverse, again from a historical perspective.
- Several sentences are grammatically unclear, and overgeneralized claims (e.g., about the historical understanding of non-coding regions) need more nuanced discussion and stronger citations.
I have added many more citations – I think the reviewer might enjoy those that give the history of the field.
- This review needs a strong conclusion section. In the last paragraph it mentions rapid revisions in genomic understanding and new technologies, however it does not clearly state what these specific revisions or technologies are, nor does it provide concrete examples of breakthroughs. Also, phrases like "these insights lead to novel strategies" and "many of these advances may be facilitated" remain abstract. What specific diseases or applications are being targeted? How do these strategies compare to existing ones?
The review summarizes the state of the art and suggests new approaches as there have been no therapeutic successes so far. The summary has been added and suggestions for future advances given.
- Also, the last section shifts focus to triplexes and therapeutic applications without clearly tying them back to flipons. If flipons are central to the review, their future research directions should be explicitly stated. A future outlook should acknowledge limitations or unanswered questions in the field, what are the biggest hurdles in applying these technologies?
I have rewritten the future directions to address these questions
Round 2
Reviewer 1 Report
Comments and Suggestions for Authors
I think that the article is now publishable. I would still use different colors or shades in Figure 3 to emphasize different strands. Similarly, in Figure 6, the strands are colored according to base pair enumeration. I would rather use a solid color per strand.
Author Response
Comment 1: I would still use different colors or shades in Figure 3 to emphasize different strands.
Thanks for the suggestion. I think the figure is better now with the different colors that highlight the different conformations
Similarly, in Figure 6, the strands are colored according to base pair enumeration. I would rather use a solid color per strand.
Many of these structures are either single-stranded or duplex, so just recoloring them does not help much. Instead, I added labels to the different structures in the RNA cloud as this will make it easier for the reader to know what they are